# The influence of social determinants of health on orthopedic trauma outcomes in the United States: A scoping review

Rachel Rumana[1]*, Hannah Mosher[1], Rebecca Landau[2], Layal Hneiny[3], Giselle Hernandez[1]

1 Department of Orthopaedic Surgery, University of Miami Miller School of Medicine, Miami, Florida, United States of America, 2 Department of Orthopaeadic Surgery, Lenox Hill Hospital at Northwell Health, New York, United States of America, 3 Louis Calder Memorial Library, University of Miami Miller School of Medicine, Miami, Florida, United States of America

* rxr1239@med.miami.edu

## Abstract

Orthopedic trauma disproportionately affects vulnerable populations in the United States, yet the influence of social determinants of health (SDOH) on patient outcomes remains underexplored. This scoping review aims to map and evaluate existing literature on the relationship between SDOH and adult orthopedic trauma outcomes in the U.S., with the goal of identifying research trends, gaps, and priorities for future investigation. Following the JBI methodology for scoping reviews and reported according to PRISMA-ScR guidelines, seven databases were systematically searched. Eligible studies included adult patients (≥18 years) treated for orthopedic trauma in U.S. trauma centers and examined outcomes in relation to SDOH domains. From 8,105 initial records, 40 studies met inclusion criteria. The majority employed retrospective designs (70%) and were conducted at single institutions (52.5%). Social and community context (80%) and healthcare access (57.5%) were the most frequently studied SDOH domains, whereas education access and quality was rarely addressed (7.5%). Lower extremity (35%) and hip (22.5%) injuries were most studied, with surgical management predominating (70%). Reported outcomes varied widely, with mortality (32.5%), length of stay (30%), infection (22.5%), and re-operation (22.5%) being most common. This review highlights both the breadth and fragmentation of current evidence, revealing significant gaps in the study of underrepresented SDOH domains and non-surgical trauma care. Future research should prioritize standardized SDOH data collection and increased investigation of non-surgical trauma outcomes in order to inform equitable clinical practice and guide policy in US orthopedic trauma care.

**Data availability statement:** All data are in the paper and the Supporting Information files.

**Funding:** The author(s) received no specific funding for this work.

**Competing interests:** The authors have declared that no competing interests exist.

## Introduction

The United States is home to a racially, ethnically, and socioeconomically diverse population; however, this diversity is not always adequately reflected in clinical research, particularly in fields like orthopedic trauma that depend heavily on equitable access to care and consistent follow-up for recovery. Numerous barriers, such as limited awareness of research opportunities, cultural stigma, and mistrust in clinical institutions, contribute to the underrepresentation of marginalized groups [1,2]. In addition to these patient-level barriers, limited access to research and training mentorship for investigators from historically underrepresented backgrounds contribute to a research workforce that does not fully reflect the diversity of the US population [3,4]. This lack of diversity influences which questions are prioritized and can reduce cultural relevance and trust during study recruitment [5]. As a result, inequitable research opportunities for diverse investigators indirectly perpetuate the under-representation of marginalized groups in orthopedic trauma studies. Together, these factors highlight the importance of examining how structural and social determinants shape participation and outcomes in orthopedic trauma research.

These disparities are closely linked to social determinants of health (SDOH), which are defined as the non-medical factors that influence health outcomes [6]. The key domains of SDOH include healthcare access and quality, neighborhood and built environment, economic stability, education access and quality, and social and community context. Each of these domains contains a range of interrelated factors that can significantly influence individual health outcomes. Healthcare access and quality involves the availability, affordability, and quality of medical services. Neighborhood and built environment encompass physical surroundings such as housing, transportation, and infrastructure. Economic stability includes income, employment, and financial security. Education access and quality refers to the availability and standard of early childhood, primary, secondary, and higher education. Social and community context includes gender, sex, race, ethnicity, social relationships, and community support systems. Many of these social determinants are deeply interconnected. For example, education, employment, and housing collectively contribute to a person's socioeconomic status. While some factors, such as race and ethnicity, are unchangeable demographic characteristics that are markers of structural and systemic inequities, others can be modified through policy and healthcare interventions.

In the context of orthopedic trauma, these social determinants play a critical role. Orthopedic trauma injuries—ranging from isolated fractures to complex, multi-system injuries—can be life-altering, requiring patients to navigate complex treatment and rehabilitation processes [7]. Effective recovery often depends on patients' ability to understand and adhere to treatment plans, which may be influenced by their educational background, economic situation, social support, and access to care [8–12]. Understanding these relationships are essential for developing effective orthopedic trauma care strategies.

Existing literature has shown that orthopedic trauma can profoundly impact a person's socioeconomic status, psychological well-being, and overall quality of life [13]. Key social

factors such as race and ethnicity, housing stability, education level, employment status, and availability of social support have been identified as influential to patient outcomes [14]. Social support—while fluid—can significantly deteriorate following a traumatic injury, potentially complicating recovery [15]. Orthopedic trauma is a broad field whereas it encompasses both the surgical and nonoperative management of injuries affecting many parts of the body including the upper and lower extremities, hip, and pelvis. While there is existing literature on specific aspects of SDOH and their impact in orthopedic trauma, there is much to still be explored.

Specifically, there is a lack of consolidated evidence examining how these factors influence adult orthopedic trauma outcomes in the United States. We restricted our review to the US to ensure that included studies reflected a consistent healthcare system and sociocultural context. Differences in healthcare delivery, access to care, and social determinants in other countries could bias comparisons, so this focus allows for a more accurate assessment of the existing literature and identification gaps relevant to US populations. In particular, previous studies have shown that the US differs from other high-income nations in geographic health disparities, income-related inequities, and how social services and healthcare are funded and delivered [16–18]. To our knowledge, no existing scoping review has addressed this intersection. The aim of this project is to map the extent, nature, and scope of existing literature on the influence of social determinants of health on orthopedic trauma outcomes in adults. Given the broad and varied nature of both orthopedic trauma and SDOH, a scoping review is the most appropriate methodology. This approach allows for the inclusion of diverse study designs and outcome measures and is well suited to identify research gaps and guide future investigations. Ultimately, this work intends to serve as a foundation for further research and contribute to efforts to advance equity in trauma care.

## Methods

A scoping review was chosen due to the broad nature of the research topic. Scoping reviews are a type of evidence synthesis reviews that present a broad overview of the evidence on a particular topic, regardless of study quality. This makes them useful in clarifying key concepts and thus identifying gaps [19]. This was conducted in accordance with the JBI methodology for scoping reviews and reported according to the PRISMA-ScR extension for scoping reviews in S1 Data [20,21]. There was no review protocol.

### Research question

What is the extent of literature in addressing how SDOH affect adult orthopedic trauma outcomes in the United States, and what should future research focus on?

### Search strategy

The search strategy was developed by an academic health science librarian (L.H.) in consultation with members of the project team (R.R. and R.L.). The search strategy was guided by CDC framework for social determinants of health, with search terms mapped to its five domains mentioned in the introduction and relevant orthopedic trauma outcomes to ensure comprehensive literature capture. The search strategy was structed around four key concepts: SDOH, orthopedic trauma injuries, outcomes such as mortality, length of stay, and post-operative complications, and US studies to ensure relevance to comparable healthcare and sociocultural contexts. This was initially done on OVID Medline and then translated using each database's syntax, search fields, and controlled vocabulary to six other databases. The controlled vocabulary across OVID Medline & Cochrane Central, Embase, Cumulative Index to Nursing and Allied Health Literature (CINAHL) Plus, SportDiscuss which are: Medical Subject Headings (MeSH), Emtree, CINAHL Headings, and SportDiscuss. We searched Ovid MEDLINE(R) ALL (United States National Library of Medicine), EMBASE (Elsevier, Embase.com), Cochrane Library (Wiley), CINAHL Plus with Full Text (Ebsco), SPORTDiscus with Full Text(Ebsco), SCOPUS (Elsevier), and Web of Science platform (Clarivate: Science Citation Index Expanded, Social Sciences Citation Index, Arts & Humanities Citation Index, Conference Proceedings Citation Index-Science, Conference Proceedings Citation Index-Social Science & Humanities, Emerging Sources Citation Index). On November 21–22, 2024, all the six databases were searched.

The records of Ovid Medline, Cochrane Library & Web of Science database records were downloaded to EndNote 20 [4] and then uploaded to Covidence web-based software. As for the remaining databases: Embase, Scopus, Plus with Full Text, and SPORTDiscus with Full Text, their records were uploaded directly to Covidence which is the tool used for deduplication, title-abstract screening, and full text evaluation [5] of all the records of the seven databases. For the full text screening, EndNote 20 was utilized to retrieve PDFs from University of Miami (UM)'s knowledgebase and then uploaded in bulk to covidence. The uncaptured records by EndNote Library were ordered through UM's Interlibrary Loan.

### Study selection

Two reviewers (R.R. and H.M.) independently reviewed all titles and abstracts for assessment against the inclusion criteria. Potentially relevant sources were moved onto full text screening, and the full text was assessed against the inclusion criteria by the same two independent reviewers. Reasons for exclusion of sources of evidence at full text were recorded and reported. Any disagreements that arose between the reviewers at any stage of the selection process were resolved by a third reviewer (R.L.).

### Inclusion and exclusion criteria

This scoping review included studies focused on orthopedic trauma outcomes—both surgical and nonoperative—in adults aged 18 and older. For the purposes of this review, orthopedic trauma was defined as acute appendicular and pelvic musculoskeletal injuries requiring or nonoperative management in adults, typically resulting from high- or low- energy trauma. This included but was not limited to fractures of the proximal humerus, clavicle, elbow, pelvis, hip, femur, patella, and tibia. Spinal, hand, and foot fractures were excluded. Only studies that addressed variations in these outcomes based on social determinants of health (SDOH)—such as race/ethnicity, marital status, housing insecurity, sex, and educational status—were included. Studies were included only if a given SDOH domain was explicitly analyzed in relation to orthopedic trauma outcomes; reporting SDOH variables in baseline demographics alone were not sufficient. The review was limited to research conducted in U.S.-based trauma centers and emergency departments to ensure consistency in sociocultural and healthcare contexts. A wide range of study designs were considered, including experimental, quasi-experimental, analytical and descriptive observational studies, case reports, and qualitative research. Systematic reviews and opinion pieces that aligned with the inclusion criteria were also included. Studies involving participants under 18 years old or non-traumatic injuries, or those conducted outside the United States, were excluded. There were no limitations for language or date.

### Data charting

Eligible studies were summarized and reported by two independent reviewers in Covidence. The following information was recorded for each article: title, author, journal, year of publication, study type, and setting (single institution, multiple institutions, regional database, national database, or other). Each study was further categorized based on orthopedic treatment as surgical, nonoperative, or combination. Due to the heterogeneity of the data, SDOH factors were categorized into the following domains: healthcare access, neighborhood and built environment, economic stability, social and community context, and education access and quality. Treatment outcome in relation to SDOH was also recorded. Because this is a scoping review, qualitative findings were summarized descriptively rather than synthesized. Counts or percentages of study characteristics, SDOH factors, and outcomes were used to map patterns across studies. All studies were weighted equally regardless of sample size.

## Results

### Study selection

Based on the PRISMA guidelines, the initial search strategy yielded 8,105 publications. After removing duplicates, 4,654 studies remained for title/abstract screening. 51 studies subsequently met inclusion criteria, with 40 studies ultimately

included after full-text review (Fig 1). Articles were excluded for various reasons including wrong setting (n = 1), wrong outcomes (n = 6), wrong study design (n = 1), and wrong patient population (n = 3). An overview of the studies included is summarized in Table 1.

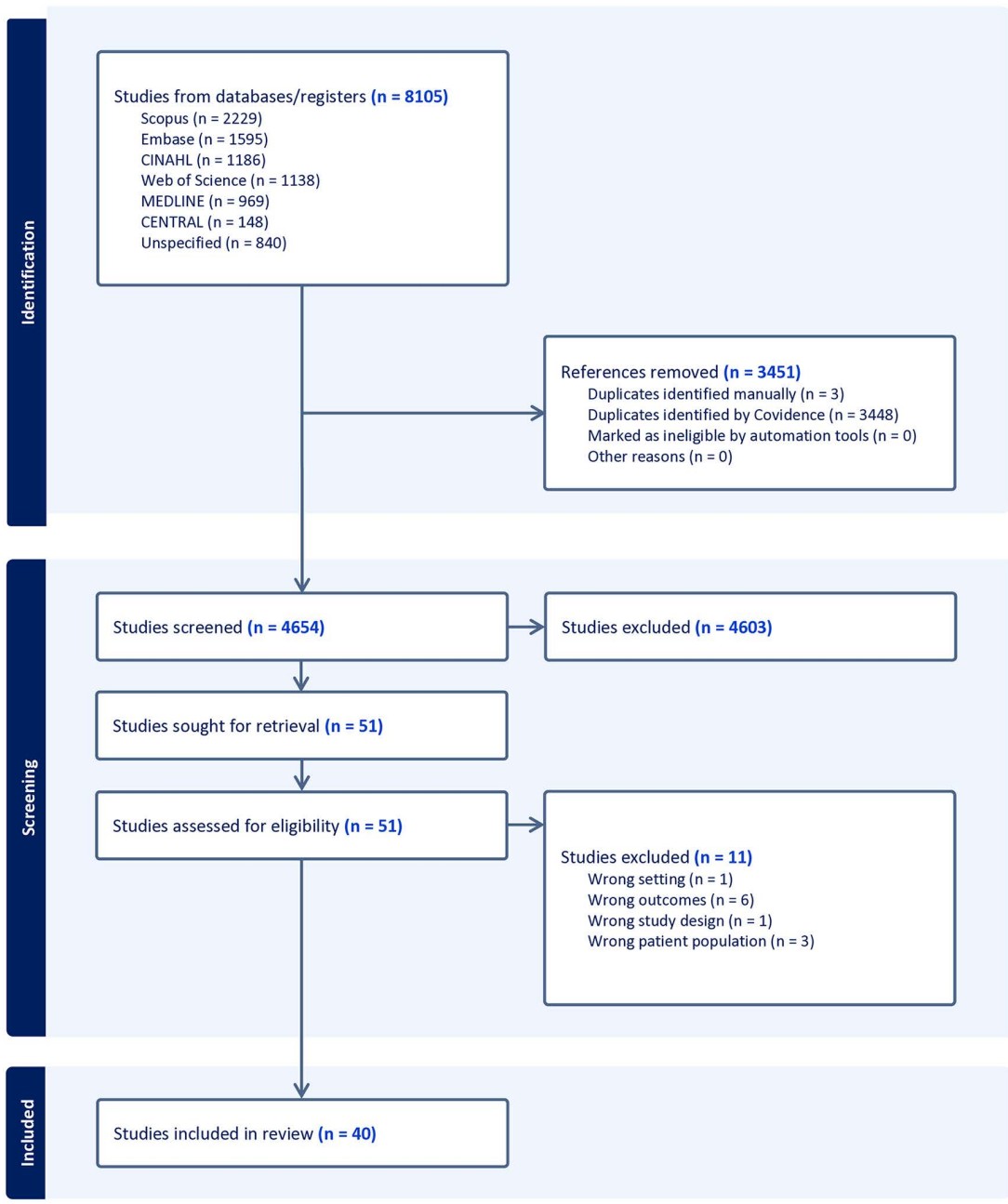

**Fig 1. PRISMA flow diagram.**

Table 1. Included studies: year, article type, journal article type, and primary outcomes.

| Author (Year) | Title | Article Type | Study Setting | Journal | Sample Size (N=) | SDOH Assessed | Outcomes Assessed |
|---|---|---|---|---|---|---|---|
| Abernathy et al (2023) [22] | Exposing the Care Conundrum of Low-Energy Pelvic Ring Fractures in Older Adults: A Review of 322 Patients | Retrospective review | Multiple Institutions | Geriatric Orthopaedic Surgery & Rehabilitation | 322 | Healthcare access; Social and community context | Long term mortality; Length of stay |
| Adams et al (2017) [23] | The Impact of Gender on Complications and Outcomes of Pelvic Fracture | Other: Brief Report | Single Institution | ACS Clinical Congress | 111 | Social and community context | Long term mortality; Length of stay; Other: Ventilator days, ICU days, pneumonia, pulmonary embolus, blood transfusion, solid organ injury, thoracic injury |
| Aizpuru et al (2018) [24] | Determinants of Length of Stay After Operative Treatment for Femur Fractures | Retrospective review | Single Institution | Journal of Orthopedic Trauma | 321 | Healthcare access; Social and community context | Length of stay |
| Bakhsh et al (2020) [25] | Ankle fractures: What role does level of insurance play in recovery and outcomes? | Retrospective cohort | Single Institution | Trauma | 209 | Healthcare access | Other: VAS pain score, PROMIS (mood, function, and pain), post-operative complications, narcotics usage |
| Bindner et al (2022) [26] | Standardized protocol for hip fracture care leads to similar short-term outcomes despite socioeconomic differences in patient populations: a retrospective cohort study | Retrospective cohort | Multiple Institutions | Current Orthopedic Practice | 125 | Healthcare access; Neighborhood and built environment; Economic stability; Social and community context | Length of stay; Other: any complication |
| Bolorunduro et al (2013) [27] | Disparities in trauma care: are fewer diagnostic tests conducted for uninsured patients with pelvic fracture? | Retrospective review | National Database | The American Journal of Surgery | 21,000 | Healthcare access | Long term mortality |
| Bontrager et al (2024) [28] | The Cost of Poverty: The Relationship Between Insurance Status, Length of Stay, and Discharge Disposition in Trauma Patients | Retrospective cohort | Single Institution | The American Surgeon | 185 | Healthcare access | Length of stay; Other: Complications |
| Branch et al (2015) [29] | Assessment of racial and sex disparities in open femoral fractures | Retrospective cohort | National Database | The American Journal of Surgery | 9,406 | Social and community context | Long term infection; Long term mortality; Other: ARDS, cardiac arrest, myocardial infarction, deep vein thrombosis, pulmonary embolism, sepsis, wound disruption, any overall complication |
| Chandrupatla et al (2024) [30] | Women undergoing primary total hip arthroplasty (THA) for hip fracture have lower in-hospital mortality compared to men | Retrospective cohort | National Database | Injury | 400,930 | Healthcare access; Neighborhood and built environment; Economic stability; Social and community context | Other: in-hospital mortality |

(Continued)

**Table 1.** (Continued)

| Author (Year) | Title | Article Type | Study Setting | Journal | Sample Size (N=) | SDOH Assessed | Outcomes Assessed |
|---|---|---|---|---|---|---|---|
| Clark et al (2005) [31] | Injuries Among Older Americans With and Without Medicare | Retrospective cohort | National Database | American Journal of Public Health | 3,100,672 | Healthcare access | Length of stay; Other: died in hospital |
| DeBaun et al (2024) [32] | Persistent racial disparities in postoperative management after tibia fracture fixation: A matched analysis of US medicaid beneficiaries | Retrospective cohort | National Database | Injury | 5,472 | Social and community context | Long term infection; Re-operation; Other: tibial delayed healing, compartment syndrome, mechanical breakage, mechanical displacement, other mechanical complications |
| Dy et al (2016) [33] | Racial and Socioeconomic Disparities in Hip Fracture Care | Retrospective cohort | Regional Database | Journal of Bone and Joint Surgery | 197,290 | Healthcare access; Neighborhood and built environment; Social and community context | Re-operation; Other: 90-day readmission, 90-day complication, 1 year in hospital mortality |
| Garlapaty et al (2024) [34] | Pre-injury methamphetamine use is associated with increased length of hospital stay in rural orthopaedic trauma patients | Retrospective cohort | Single Institution | Injury | 249 | Healthcare access; Neighborhood and built environment; Social and community context | Length of stay |
| Hartline et al (2024) [10] | Socioeconomic status is associated with greater hazard of post-discharge mortality than race, gender, and ballistic injury mechanism in a young, healthy, orthopedic trauma population | Retrospective cohort | Single Institution | Injury | 2,539 | Healthcare access; Neighborhood and built environment; Social and community context | Long term mortality |
| Henstenburg et al (2021) [35] | Risk factors for complications following pelvic ring and acetabular fractures: A retrospective analysis at an urban level I trauma center | Retrospective review | Single Institution | Journal of Orthopaedics, Trauma, and Rehabilitation | 126 | Healthcare access | Long term infection; Length of stay; Re-operation |
| Hong et al (2022) [36] | The Effect of Social Deprivation on Fracture-Healing and Patient-Reported Outcomes Following Intramedullary Nailing of Tibial Shaft Fractures | Retrospective cohort | Single Institution | The Journal of Bone and Joint Surgery | 229 | Neighborhood and built environment | 30 day infection; Re-operation; Other: PROMIS and RUST |
| Jarman et al (2021) [37] | The Impact of Delayed Management of Fall-Related Hip Fracture Management on Health Outcomes fro African American Older Adults | Retrospective review | National Database | Journal of Trauma Acute Care Surgery | 199,870 | Social and community context | Length of stay; Other: catheter associated urinary tract infection (CAUTI), deep vein thrombosis (DVT), decubitus ulcer, and pulmonary embolism |
| Kay et al (2014) [38] | The Homeless Orthopaedic Trauma Patient: Follow-up, Emergency Room Usage, and Complications | Retrospective review | Single Institution | Journal of Orthopedic Trauma | 126 | Economic stability | Long term infection; Other: hardware failure and nonunion |

*(Continued)*

| Author (Year) | Title | Article Type | Study Setting | Journal | Sample Size (N=) | SDOH Assessed | Outcomes Assessed |
|---|---|---|---|---|---|---|---|
| Kiel et al (1994) [39] | The Outcomes of Patients Newly Admitted to Nursing Homes after Hip Fracture | Retrospective cohort | Multiple Institutions | American Journal of Public Health | 2,624 | Social and community context | 30 day mortality; Other: rehospitalization |
| Low et al (2017) [40] | Complications and revision amputation following trauma-related lower limb loss | Retrospective cohort | National Database | Injury | 2,879 | Social and community context | Length of stay; Re-operation; Other: major post-surgical complication |
| MacKenzie et al (2005) [41] | Long-Term Persistence of Disability Following Severe Lower-Limb Trauma | Other: Prognostic | Single Institution | The Journal of Bone and Joint Surgery | 569 | Healthcare access; Economic stability; Social and community context; Education access and quality | Other: Physical and Psycho-social SIP Subscores |
| Mandl et al (2024) [42] | The Effect of Social Isolation on 1-Year Outcomes After Surgical Repair of Low-Energy Hip Fracture | Prospective | Single Institution | Journal of Orthopaedic Trauma | 325 | Healthcare access; Neighborhood and built environment; Social and community context; Education access and quality | Other: PROMIS-29, LEAS |
| Mathew et al (2015) [43] | All pelvises are created equal: or are they? Sex differences in pelvic trauma | Abstract only | Single Institution | American College of Surgeons | 382 | Social and community context | Long term mortality |
| Mundy et al (2023) [44] | Does treatment at a level I trauma center reduce disparities in patient outcomes for open tibia fractures? A retrospective analysis of the National trauma Databank | Retrospective cohort | National Database | Journal of Clinical Orthopaedics and Trauma | 81,855 | Healthcare access; Social and community context | Length of stay; Other: injury-specific complications, specific complications |
| Okike et al (2018) [45] | Association Between Race and Ethnicity and Hip Fracture Outcomes in a Universally Insured Population | Retrospective review | Regional Database | The Journal of Bone and Joint Surgery | 17,790 | Social and community context | Long term mortality; Re-operation; ED visit; Other: 90-day postoperative complications, in-hospital decubitus ulcers, 90-day unplanned readmissions |
| Patel et al (2023) [46] | Increased Neighborhood Deprivation Is Associated with Prolonged Hospital Stays After Surgical Fixation of Traumatic Pelvic Ring Injuries | Retrospective cohort | Single Institution | Journal of Bone and Joint Surgery | 134 | Healthcare access; Neighborhood and built environment; Economic stability; Social and community context; Education access and quality | Length of stay |

*(Continued)*

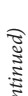

Table 1. (Continued)

| Author (Year) | Title | Article Type | Study Setting | Journal | Sample Size (N=) | SDOH Assessed | Outcomes Assessed |
|---|---|---|---|---|---|---|---|
| Patel et al (2024) [47] | Greater Socioeconomic Deprivation Is Associated With Increased Complication Rates and Lower Patient-Reported Outcomes Following Open Reduction and Internal Fixation of Humeral Shaft Fractures | Retrospective cohort | Single Institution | Journal of Orthopaedic Trauma | 196 | Neighborhood and built environment; Economic stability; Social and community context | Other: blood tranfusions, PROMIS, overall complication rate, nonunion, radial nerve injury, shoulder dysfunction, time to radiographic union, elbow arc of motion |
| Petrigliano et al (2012) [48] | Factors Predicting Complication and Reoperation Rates Following Surgical Fixation of Proximal Humeral Fractures | Retrospective cohort | Regional Database | Journal of Bone and Joint Surgery | 11,394 | Neighborhood and built environment; Economic stability; Social and community context | 30 day mortality; Re-operation; Other: pulmonary embolism, neurovascular injury |
| Piposar et al (2012) [49] | Race May Not Effect Outcomes in Operatively Treated Tibia Fractures | Retrospective cohort | Single Institution | Clinical Orthopaedics and Related Research | 302 | Social and community context | 30 day mortality; Long term infection; Length of stay; Re-operation; Other: compartment syndrome, deep vein thrombosis, pulmonary embolism |
| Prather et al (2020) [50] | Factors affecting emergency department visits, readmissions, and reoperations within 30 days of ankle fracture surgery- an institutional retrospective study | Retrospective cohort | Single Institution | Injury | 596 | Healthcare access; Social and community context | Re-operation; ED visit; Other: re-admission |
| RafaelArceo et al (2018) [51] | Disparities in follow-up care for ballistic and non-ballistic long bone lower extremity fractures | Retrospective cohort | Single Institution | Injury | 612 | Economic stability; Social and community context | ED visit |
| Richardson et al (2013) [52] | Fee-based Care is Important for Access to Prompt Treatment of Hip Fractures Among Veterans | Retrospective cohort | Regional Database | Clinical Orthopaedics and Related Research | 11,189 | Healthcare access; Neighborhood and built environment; Social and community context | Long term mortality |
| Roberts et al (2024) [53] | The Effect of Age and Sex on Early Postoperative Outcomes after Surgical Treatment of Distal Radius Fractures | Retrospective review | Single Institution | The Journal of Hand Surgery | 521 | Social and community context | Other: postoperative complications |
| Solasz et al (2023) [54] | Factors Associated With the Development of a Confirmed Fracture-Related Infection | Retrospective cohort | Multiple Institutions | Journal of Orthopaedic Trauma | 193 | Social and community context | 30 day infection |
| Thomas et al (2023) [55] | The Role of Geographic Disparities in Outcomes after Orthopaedic Trauma Surgery | Retrospective cohort | National Database | Injury | 235,393 | Neighborhood and built environment; Economic stability; Social and community context | Long term mortality; Other: overall complication |

(Continued)

Table 1. (Continued)

| Author (Year) | Title | Article Type | Study Setting | Journal | Sample Size (N=) | SDOH Assessed | Outcomes Assessed |
|---|---|---|---|---|---|---|---|
| Truong et al (2024) [56] | The Association Between Social Determinants of Health and Distal Radius Fracture Outcomes | Retrospective cohort | National Database | Journal of Hand Surgery | 114,050 | Healthcare access; Economic stability; Social and community context | Long term infection; Re-operation; ED visit; Other: wound dehiscence, nerve injury or CTR, CRPS, malunion/nonunion |
| Verlinsky et al (2022) [57] | Socioeconomic status does not change decision-making in the treatment of distal radius fractures at a level 1 trauma center | Retrospective cohort | Single Institution | Orthopaedic Trauma International | 816 | Healthcare access; Economic stability | Length of stay |
| Wheelwright et al (2022) [58] | High Performance Outcome After Tibial Plafond Fracture: The Significant Factors | Retrospective cohort | Single Institution | Foot and Ankle International | 198 | Healthcare access; Neighborhood and built environment; Social and community context | Other: PROMIS |
| Worden et al (2015) [59] | Does insurance status alter short-term outcomes if a standardized fracture treatment pathway is utilized? | Retrospective cohort | Multiple Institutions | Current Orthopaedic Practice | 204 | Healthcare access; Social and community context | Length of stay; Other: complications |
| Zelle et al (2021) [60] | Fate of the Uninsured Ankle Fracture: Significant Delays in Treatment Result in an Increased Risk of Surgical Site Infection | Retrospective cohort | Single Institution | Journal of Orthopaedic Trauma | 489 | Healthcare access; Social and community context | Long term infection |
| TOTAL: Count (%) | | Retrospective cohort - 28 (70); Retrospective review – 8 (20); Prospective – 1 (2.5); Brief Report – 1 (2.5); Abstract only – 1 (2.5); Prognostic – 1 (2.5) | Single – 21 (52.5); National Database – 10 (25); Multiple Institutions – 5 (12.5); Regional Database – 4 (10%) | | | Social and Community Context – 32 (80); Healthcare Access – 23 (57.5); Neighborhood and Built Environment – 13 (32.5); Economic Stability – 11 (27.5); Education Access and Quality – 3 (7.5) | |

## Research methodology and study setting

As shown in Table 1, most studies were retrospective in nature, with 28 (70%) employing a retrospective cohort design and 8 (20%) categorized as retrospective reviews. Only a small proportion of studies used prospective (2.5%) or prognostic (2.5%) designs, with brief reports and abstract-only publications each accounting for (2.5%) of the total. Regarding study setting (Table 1), most studies were conducted at a single institution (52.5%), followed by those using national databases (25%). Fewer studies involved multiple institutions (12.5%) or regional databases (10%).

## SDOH domains

As presented in Table 1, Social and Community Context was the most frequently referenced social determinant of health (SDOH), mentioned in 32 studies (80%). Healthcare Access appeared in 23 studies (57.5%), followed by Neighborhood and Built Environment in 13 studies (32.5%) and Economic Stability in 11 studies (27.5%). Education Access and Quality was the least represented domain, cited in only 3 studies (7.5%).

## Orthopedic trauma injury – location and treatment

Table 2 shows the most studied injury locations were the lower extremity (35%) and hip (22.5%), followed by the pelvis (17.5%). Upper extremity and other injury locations each accounted for 12.5% of the studies. As shown in Table 3, surgical treatment was the predominant management approach, reported in 28 studies (70%). A combination of surgical and non-surgical treatments were described in 9 studies (22.5%), while non-surgical treatment alone was reported in just 1 study (2.5%). In 2 studies (5%), the treatment type was not specified.

## Treatment outcomes

As shown in Table 4, the most reported treatment outcomes were categorized as "other," appearing in 27 studies (67.5%). All outcomes grouped under "other" are detailed in Table 1, which provides a comprehensive summary of each study's reported measures. Mortality was reported in 13 studies (32.5%), followed closely by length of stay in 12 studies (30%). Infection and re-operation were each mentioned in 9 studies (22.5%), while emergency department visits were the least frequently reported outcome, cited in 4 studies (10%).

**Table 2. Orthopedic injury location.**

| Location | Count (%) |
|---|---|
| Lower Extremity | 14 (35%) |
| Hip | 9 (22.5%) |
| Pelvis | 7 (17.5%) |
| Upper Extremity | 5 (12.5%) |
| Multiple | 5 (12.5%) |

**Table 3. Treatment overview.**

| Treatment | Count (%) |
|---|---|
| Surgical | 28 (70%) |
| Combination | 9 (22.5%) |
| Not Specified | 2 (5%) |
| Nonoperative | 1 (2.5%) |

PLOS Global Public Health

**Table 4. Treatment outcomes (total number of articles).**

| Treatment Outcomes | Count (%) |
|---|---|
| Other | 27 (67.5%) |
| Mortality | 13 (32.5%) |
| Length of Stay | 12 (30%) |
| Infection | 9 (22.5%) |
| Re-Operation | 9 (22.5%) |
| ED Visit | 4 (10%) |

## Discussion

This scoping review highlights both the breadth and the limitations in the investigation of orthopedic trauma outcomes in the context of social determinants of health in the United States. For the SDOH domains included in this study, social and community context was the most investigated, followed in order of decreasing frequency by healthcare access, neighborhood and built environment, economic stability, and education access and quality. This result is not surprising, as these variables are commonly included in chart reviews, such as gender, race, and ethnicity, which fall under the domain of social and community context. Healthcare access includes insurance status, which is another variable that can easily be pulled from a patient's medical chart. The other domains require researchers to retrieve the information from the patients themselves, which adds a level of difficulty and may prevent researchers from including such domains as variables of interest. The introduction of standardized data collection of SDOH-related information could allow for these aspects to be studied more systematically. It is also possible that other factors, such as lack of funding and prioritization, limited interdisciplinary collaboration, and methodological complexity, contribute to the disparity in domain count. These features would require more complicated solutions, such as creating structured interdisciplinary partnerships and advocating for targeted funding.

Orthopedic trauma is a very large sub-specialty that includes many different orthopedic injuries and treatment options. In this review, we determined that the most popular injury location was lower extremity, followed by hip, pelvis, upper extremity, and multiple locations. We also determined that the included papers focused mainly on surgical treatment options, with fewer papers including combination of surgical and non-surgical treatment options and only one paper focusing on non-surgical treatments. As highlighted by these results, the current body of literature is spread over many different injuries and treatments. Further, each injury type has differences in surgical and nonoperative treatment options available. Thus, it is difficult to group these studies and thus their outcomes by generalized location or treatment options. This spread in data does reveal, however, that there are still vast possibilities for future research on orthopedic trauma conditions and treatment options, specifically in pelvic and upper extremity injuries and non-surgical treatments.

Over 30 orthopedic trauma treatment outcomes viewed in relation to SDOH were pulled from the included studies. The most common outcomes (mortality, length of stay, infection, re-operation, and following ED visit) were included in the Covidence data charting form. Due to a large variability in time since surgery for the outcomes mortality and infection rates, time was not included in Table 4. 67.5% of the studies also included other outcomes, such as overall complications, deep vein thrombosis, hardware failure and nonunion, and delayed healing. The variety in reported outcomes would make any systemic conclusion on the included studies difficult. Future research could focus on any outcome summarized in Table 1, especially ED visits and any outcome included in others.

Although the existing evidence remains limited and heterogenous, several practical implications emerge from this review for the US. Trauma centers could integrate SDOH factors, such as insurance status and race, into discharge planning to identify patients who may require enhanced care coordination or transportation assistance. SDOH data could also inform risk stratification models to identify patients at higher risk for complications or loss to follow-up, allowing trauma

programs to allocate resources more efficiently. At the policy level, these findings highlight the importance of supporting reimbursement models and community-based partnerships that facilitate the integration of social services into trauma care. Strengthening both the clinical and policy pathways may help mitigate disparities and improve recovery trajectories for socially vulnerable patients.

This study has several limitations. The wide heterogeneity and broad scope of the studies included prevent this review from including a detailed analysis and answering specific clinical or policy questions. This also prohibits the paper from including a meta-analysis or other quantitative synthesis. The study design also does not allow for the quality of the studies to be investigated. A large proportion of the studies were retrospective and single center, which limits their generalizability. Additionally, SDOH variables were inconsistently defined and frequently obtained through chart review, relying on proxies such as insurance status or race rather than standardized measures. The methodological constraints within the existing evidence base should be considered when interpreting the findings and highlight the need for future prospective, multicenter studies with uniform SDOH data collection. Furthermore, publication bias limits the studies that were accepted to journals and made readily available in the databases. Additionally, a formal protocol was not registered prior to conducting this review, which we acknowledge as a limitation. We have tried to decrease the effects of this limitation by providing the full search strategy in S1 File to support transparency and reproducibility. Finally, this scoping review is limited to studies inside of the US, which limits the generalizability worldwide We believe that this decrease in generalizability is warranted, as including studies not based in the US would introduce biases due to differences in healthcare systems and sociocultural contexts. Even with these limitations, we believe that this study is a valuable exploration into the current state of literature on how orthopedic trauma outcomes are affected by SDOH.

## Conclusion

This scoping review identifies a growing but still fragmented body of literature examining how social determinants of health influence orthopedic trauma outcomes in the United States. The predominance of retrospective studies focused on limited SDOH domains, specifically social and community context and healthcare access, highlights both the feasibility and constraints of relying on existing clinical data. Significant gaps remain in evaluating the other domains, especially education access and quality. Additionally, drawing systematic conclusions is prevented by the wide variability in orthopedic trauma injuries, treatment options, and reported outcomes. These findings underscore the need for prospective, multidisciplinary research that incorporates consistent SDOH data collection and prioritizes health equity in orthopedic trauma care. By addressing these gaps, future work can help inform targeted interventions and policies to improve outcomes for vulnerable populations affected by orthopedic trauma.

## Supporting information

**S1 Data. PRISMA-ScR Checklist.**
(DOCX)

**S1 File. Full Search Strategy.**
(XLSX)

## Author contributions

**Conceptualization:** Rachel Rumana, Hannah Mosher, Rebecca Landau, Layal Hneiny, Giselle Hernandez.

**Data curation:** Rachel Rumana, Layal Hneiny.

**Formal analysis:** Rachel Rumana.

**Investigation:** Rachel Rumana, Hannah Mosher, Layal Hneiny.

**Methodology:** Rachel Rumana, Rebecca Landau.

**Project administration:** Giselle Hernandez.

**Resources:** Layal Hneiny.

**Supervision:** Rachel Rumana.

**Writing – original draft:** Rachel Rumana, Hannah Mosher.

**Writing – review & editing:** Rachel Rumana, Hannah Mosher, Rebecca Landau, Layal Hneiny, Giselle Hernandez.

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
