## [Decision Letter · Decision Letter 0]

14 Oct 2025

PGPH-D-25-02410

The Influence of Social Determinants of Health on Orthopedic Trauma Outcomes in the United States: A Scoping Review

Dear authors

Thank you for submitting your manuscript to PLOS Global Public Health. After careful consideration, we feel that it has merit but does not fully meet PLOS Global Public Health’s publication criteria as it currently stands. Therefore, we invite you to submit a revised version of the manuscript that addresses the points raised 

Please submit your revised manuscript within 3 weeks. If you will need more time than this to complete your revisions, please reply to this message or contact the journal office at globalpubhealth@plos.org. Please include the following items when submitting your revised manuscript:

We look forward to receiving your revised manuscript.

Kind regards,

Andreas K Demetriades, MBBChir, MPhil, FRCSEd, FEBNS.

Academic Editor

Journal Requirements:

1. Please provide a/amend your detailed Financial Disclosure statement. This is published with the article. It must therefore be completed in full sentences and contain the exact wording you wish to be published.

State what role the funders took in the study. If the funders had no role in your study, please state: “The funders had no role in study design, data collection and analysis, decision to publish, or preparation of the manuscript.”

3. We have amended your Competing Interest statement to comply with journal style. We kindly ask that you double check the statement and let us know if anything is incorrect.

Additional Editor Comments (if provided):

Reviewers' comments:

Reviewer's Responses to Questions

**Comments to the Author**

1. Does this manuscript meet PLOS Global Public Health’s publication criteria?

Reviewer #1: Partly

Reviewer #2: Yes

Reviewer #3: Yes

2. Has the statistical analysis been performed appropriately and rigorously?

Reviewer #1: I don't know

Reviewer #2: N/A

Reviewer #3: Yes

3. Have the authors made all data underlying the findings in their manuscript fully available (please refer to the Data Availability Statement at the start of the manuscript PDF file)?

Reviewer #1: Yes

Reviewer #2: Yes

Reviewer #3: Yes

4. Is the manuscript presented in an intelligible fashion and written in standard English?

Reviewer #1: Yes

Reviewer #2: Yes

Reviewer #3: Yes

Reviewer #1: Introduction:

The authors are not explicit about how limited awareness of research opportunities for researchers of certain backgrounds leads to under-representation of people of those backgrounds in studies. I assume the authors are alluding to these researchers being more keen to perform studies that affect people from their community and also more likely to get buy-in and recruitment.

There is no referencing of the multiple assertions made in the introduction between SDOH domains and the orthopaedic trauma outcomes. Please find references to support your views

Methodology:

Why were distal humerus (unless this is what elbow meant), radial, and ulnar fractures not included for these limb fractures?

I note that no spinal fractures, hand fractures or foot fractures were included. How do the authors define orthopaedic trauma?

Results:

Table 2 content could be column 3 of table 1

Table 3 could be an additional column for table 1

Table 4 could be column 6 of table 1

It’s unclear why lower extremity fractures are separate from hip fractures, as I suspect you’re using hips to mean proximal femur fractures

It’s unclear how the qualitative data collected as per the methodology has been synthesised

The results do not seem to relate SDOH domains with outcomes, which was the thrust of the rationale for performing this study.

Reviewer #2: Thank you for your effort to conduct a timely and well-executed scoping review examining the influence of social determinants of health on orthopedic trauma outcomes in the United States. I would like give some recommendations:

1. It might be helpful to elaborate on the framework for SDOH and trauma outcomes that lead to your search strategy to make sure it is comprehensive.

2. It might be helpful to mention key domains/words of SDOH and trauma outcomes in the search strategy section of the main text. I mean a summary of S2 file.

3. Please elaborate why and how your restrict your review scope to "the United States"

4. It might be helpful to report on key findings across studies on "The Influence of Social Determinants of Health on Orthopedic Trauma Outcomes in the United States". Right now, it seems you are just counting the frequencies of SDOH and trauma outcomes across studies. How about the combination of these two?

Reviewer #3: Overall, this manuscript adds clarity on consideration for the social determinants of health in orthopedic trauma literature and is structured well with JBI methodology for scoping reviews and PRISMA-ScR reporting guidelines.

However, the manuscript would benefit from several revisions to strengthen transparency, interpretation, and methodological clarity.

Major Comments

1. Protocol Registration.

The manuscript notes that no protocol was registered. While not strictly required for scoping reviews, protocol registration (e.g., with Open Science Framework as PROSPERO does not permit registration of scoping reviews) enhances transparency and credibility. The authors workaround this adequately by uploading their full search strategy as supplemental information. However, the authors should also more explicitly explain the reasoning behind their decision to not register their initial protocol.

Recommendation: Acknowledge this limitation explicitly in the methods or discussion and explain why a protocol was not registered.

2. Clarification of Statistical/Methodological Approach.

The manuscript frequently reports proportions of studies that “referenced” a given SDOH domain (e.g., 80% included social/community context, 7.5% included education). However, it is not entirely clear what constitutes a “reference.” For example: Was an SDOH domain counted if mentioned in baseline demographics only? Or did the study need to analyze the SDOH factor as a predictor of outcomes? Were studies weighted equally regardless of sample size?

Recommendation: Provide more explicit criteria for how studies were coded as including a particular SDOH domain. Clarify whether this reflects analysis of SDOH as a variable in statistical analysis, mention in the text, or categorization of the study’s primary aim. This distinction is critical for interpretation, as a mere mention in baseline demographic data is not equivalent to rigorous SDOH analysis.

3. Study Quality Considerations

While quality assessment is not required in scoping reviews, the predominance of retrospective and single-center studies (70% and 52.5%, respectively) as reported in this analysis substantially limits generalizability.

Recommendation: Add a short subsection in the discussion that comments on methodological limitations of the included literature (e.g., retrospective design, lack of standardized SDOH data collection, reliance on chart review variables such as insurance or race). Even in the absence of a formal assessment of bias and quality of each source, the data the authors have reported regarding study type merits mention in the limitations.

4. Outcome Categorization

Table 7 reports that 67.5% of studies included outcomes in the category “other,” which obscures important heterogeneity especially given that this category is the largest of the table.

Recommendation: Consider listing the specific outcomes included under “other” whether in an appendix to the table or in the text of the results section itself. This would allow readers to fully appreciate the breadth of reported measures.

5. Interpretation and Clinical/Policy Implications

The conclusions appropriately emphasize research gaps, but the implications for current clinical practice and health policy are underdeveloped. For example: How could trauma centers use SDOH data in discharge planning? How might insurance status and neighborhood deprivation guide targeted follow-up or resource allocation?

Recommendation: Expand discussion on practical applications of current evidence, even if limited, to strengthen relevance for clinicians and policymakers.

6. U.S.-Specific Context

Because the review is limited to U.S. studies, it would be useful to highlight how U.S. health system characteristics (insurance-based access, trauma system organization) shape findings and limit generalizability internationally.

Recommendation: Elaborate on the US-specific context in both the limitation and discussion sections to ensure acknowledgement of limited generalizability as well as make the discussion section more relevant for US-based readers and future research.

Minor Comments

1. Abstract – Strengthen the concluding sentence by explicitly stating the main implication (e.g., “highlighting the need for standardized SDOH data collection and greater focus on non-surgical trauma care”).

2. Tables – The authors could consider combining tables 5 and 6 (injury location and treatment overview) into a unifying figure or condensing to improve readability.

3. Terminology – Race and ethnicity are categorized under “social and community context,” but this may benefit from clarification. These variables encompass both a demographic categorization as well as systemic inequities; precise terminology would strengthen the manuscript.

**Do you want your identity to be public for this peer review?** For information about this choice, including consent withdrawal, please see our Privacy Policy

Reviewer #1: No

Reviewer #2: **Yes: ** Anh Nguyen

Reviewer #3: **Yes: ** Parker Frankiewicz

---

## [Decision Letter · Decision Letter 1]

11 Dec 2025

The Influence of Social Determinants of Health on Orthopedic Trauma Outcomes in the United States: A Scoping Review

PGPH-D-25-02410R1

Dear Ms. Rumana,

We are pleased to inform you that your manuscript 'The Influence of Social Determinants of Health on Orthopedic Trauma Outcomes in the United States: A Scoping Review' has been provisionally accepted for publication in PLOS Global Public Health.

Best regards,

Julia Robinson

Executive Editor

Reviewer Comments (if any, and for reference):

Reviewer's Responses to Questions

**Comments to the Author**

Reviewer #2: All comments have been addressed

Reviewer #3: All comments have been addressed

publication criteria?

Reviewer #2: Yes

Reviewer #3: Yes

3. Has the statistical analysis been performed appropriately and rigorously?

Reviewer #2: N/A

Reviewer #3: Yes

4. Have the authors made all data underlying the findings in their manuscript fully available (please refer to the Data Availability Statement at the start of the manuscript PDF file)?

Reviewer #2: Yes

Reviewer #3: Yes

5. Is the manuscript presented in an intelligible fashion and written in standard English?

Reviewer #2: Yes

Reviewer #3: Yes

Reviewer #2: The authors have addressed well reviewers' comments

Reviewer #3: (No Response)

**Do you want your identity to be public for this peer review?** For information about this choice, including consent withdrawal, please see our Privacy Policy

Reviewer #2: **Yes: ** Anh Nguyen

Reviewer #3: **Yes: ** Parker Frankiewicz
